# Developmental Disorder Probability Scores at 6–18 Years Old in Relation to In-Utero/Peripartum Antiretroviral Drug Exposure among Ugandan Children

**DOI:** 10.3390/ijerph19063725

**Published:** 2022-03-21

**Authors:** Jorem Emmillian Awadu, Alla Sikorskii, Sarah Zalwango, Audrey Coventry, Bruno Giordani, Amara E. Ezeamama

**Affiliations:** 1Department of Psychiatry, Michigan State University, East Lansing, MI 48824, USA; awadujor@msu.edu (J.E.A.); sikorska@msu.edu (A.S.); 2Directorate of Public Health and Environment, Kampala Capital City Authority, Kampala 00256, Uganda; skzalwango@gmail.com; 3Spectrum Health Incorporated, Lansing, MI 49085, USA; audrey.coventry@spectrumhealth.org; 4Department of Psychiatry, University of Michigan, Ann Arbor, MI 48109, USA; giordani@med.umich.edu

**Keywords:** peripartum ART exposure, HIV-exposed uninfected, perinatal HIV infection, ASD, ADHD, functional impairment, resiliency, children, adolescents, Uganda

## Abstract

(1) We examined the hypothesis that in utero/peripartum antiretroviral (IPA) exposure may affect the likelihood of developmental disorders—i.e., attention deficit and hyperactivity disorder (ADHD), autism spectrum disorder (ASD), and functional impairment (FI). (2) Children and their primary caregivers were enrolled and followed for 12 months. The sample included 250 children perinatally HIV-infected (CPHIV), 250 children HIV-exposed and uninfected (CHEU) of women living with HIV, and 250 children HIV unexposed and uninfected (CHUU) at 6–18 years of age. CHEU’s IPA exposure -type was established via medical records and categorized as no IPA, single-dose nevirapine with/without zidovudine (SdNVP ± AZT), SdNVP + AZT + Lamivudine (3TC), or combination ART (cART). Developmental disorders were assessed at months 0, 6, and 12 per caregiver response to standardized questions from the third edition of *Behavioral Assessment System for Children*. Multivariable repeated measures linear regression models estimated standardized mean differences (SMDs) with 95% confidence intervals (95% CI) according to the IPA exposure type relative to CHUU with adjustment for the dyad’s sociodemographic and psychosocial factors. (3) Relative to the CHUU, outcomes were similar for CPHIV/CHEU with cART, SdNVP ± AZT, and no anti-retroviral drug exposure in the peripartum period. For CHEU relative to CHUU, SdNVP + AZT + 3TC exposure was associated with lower resiliency (SMD = −0.26, 95% CI: −0.49, −0.51), and elevated scores on ADHD (SMD = 0.41, 95% CI: 0.12, 0.70), ASD (SMD = 0.40, 95% CI: 0.19, 0.61), and EBD (SMD = 0.32, 95% CI: 0.08, 0.56) probability and functional impairment (SMD = 0.39, 95% CI: 0.18, 0.61) index scores. With the exception of ADHD, the adverse association between SdNVP + AZT + 3TC and outcomes were replicated for CPHIV vs. CHUU. (4) The results provided reassuring evidence that cART exposure in the peripartum period is unlikely to be adversely associated with developmental disorder probability scores in late childhood and adolescent years. However, the peripartum SdNVP + AZT + 3TC exposure associated elevation in developmental disorder probability and functional limitation at 6–18 years of life is a concern.

## 1. Introduction

Between 2000 and 2018, an estimated 15 million children were born to pregnant women living with HIV (PWLWH) [1]. Out of necessity, many of these HIV-exposed children are concurrently exposed to combination antiretroviral therapy (cART) to prevent mother-to-child transmission (PMTCT) of HIV [2]. Interventions to prevent vertical transmission of HIV began in 1994 with intrapartum single dose nevirapine (SdNVP) as monotherapy. This intervention became more complex over time to include Zidovudine (AZT), Lamivudine (3TC), and eventually maternal cART as the most effective PMTCT approach. The result was a dramatic reduction in the global rate of vertical transmission to less than 5% with variations ranging from 2% to 28% among the 21 priority countries with generalized HIV epidemic [3]. In the current era, most infants born to PWLWH will be children HIV-exposed but uninfected (CHEU) [1,3]. For children perinatally HIV-infected (CPHIV), timely initiation on combination ART (cART) is essential for thriving physically, cognitively, and mentally [4,5].

Some of the ART regimens used for PMTCT in PWLWH have been associated with developmental and neurological problems, and thus were reclassified from preferred to alternative in certain contexts depending on drug interaction concerns by the Health and Human Services (HHS), World Health Organization (WHO), or other such regulatory agencies [6]. For example, nucleoside reverse transcriptase inhibitors (NRTIs)—a first generation of drug that was used for PMTCT—have been associated with acquired mitochondrial toxicity which contributes to DNA alteration and increases the likelihood of mental health developmental disorders [7]. Impaired cognition/neurodevelopment, as well as emotional and behavioral problems are common comorbid conditions in children with developmental disorders with onward adverse physical and mental health impacts throughout the life course [8,9,10].

Risk factors for neurodevelopmental disorders in the general population include prematurity, low birth weight, high psychosocial stress, poverty, encephalopathy, substance abuse, and stunting [11]. These generally occur at higher frequency in HIV-affected populations [5,12,13,14,15]. The vulnerability of CHEU and CPHIV to adverse neurocognitive and behavioral outcomes is an emergent public health concern [16,17,18,19]. However, few comparative long-term pharmacovigilance studies have specifically investigated the relationship of prenatal ART exposure to developmental disorders [20,21]. In the African setting, investigation of developmental disorders has often focused on ADHD with an estimated prevalence rate among CPHIV of 6% in Uganda [19], 12% in Kenya [22], and 17% in South Africa [23]. These can be compared to an estimated general population global ADHD prevalence rate of 5% [24], 11% ADHD prevalence among CHEU, and 12% ADHD prevalence rate among CPHIV from the United States [18]. Information from African settings are limited and, where available, often does not include appropriate comparators of CHUU and CHEU. However, available information suggests that certain developmental disorders may be higher in HIV-affected populations. Furthermore, due to resource limitations for detection and diagnosis, the prevalence of certain developmental disorders may be underestimated in resource-limited settings such as Uganda. Hence, we inform the existing gaps in knowledge by investigating the relationship of early ART with developmental disorders, functional impairment, and resiliency among Ugandan HIV-affected children and CHUU. We hypothesized that the probability index scores for developmental disorder, functional impairment, and resiliency outcomes at 6–18 years of life would be worse in CHEU and CPHIV relative to CHUU. We further hypothesized that the probability index scores for respective outcomes would differ according to in utero/peripartum ART exposure type in children of PWLWH.

## 2. Methods

### 2.1. Study Design, Participants and Study Setting

Ugandan children aged 6–18 years and their primary adult caregivers were enrolled on a first-come, first-served basis as part of two prospective cohort studies implemented between 16 March 2017 and 30 June 2021. By design, an equal number n = 250 of CPHIV, CHEU, and CHUU were recruited. CPHIV were enrolled from among patients cared for in a community health center in urban Kampala, Uganda. CHEU were identified from the early infant diagnosis (EID) registers at the same clinic. CHUU were recruited by referrals using the social networks of the child-caregiver dyads already enrolled. Current HIV status for both the CHEU and CHUU was ascertained using an HIV-rapid diagnosis test at enrolment. Each child-caregiver pair was followed for 12 months or until loss to follow-up with study related assessments at enrolment, 6, and 12 months.

### 2.2. Eligibility/Exclusion Criteria

Eligible children were between 6 and 18 years old at enrolment with a consenting adult (i.e., age ≥ 18 years) caregiver who had served in the caregiving role for at least six months before study enrolment. To ensure medical record documentation of child ART exposure status during pregnancy, their mother’s HIV status, mother’s participation in the PMTCT program, and the child’s HIV status at birth, only children born in a hospital or medical setting were enrolled. In addition to maternal ART exposure type, the child’s birth weight and prematurity status were abstracted from the delivery records. Children not born in a clinical setting and child-caregiver dyads without antenatal, delivery, or care records were ineligible because peripartum ART exposure and the HIV status of their pregnancy could not be ascertained.

### 2.3. Ethical Approval

Protocol for their respective cohort studies were reviewed by professionals from the research ethics boards of Michigan State University (IRB Protocol numbers: 16-828 and 205), Makerere University College of Health Sciences, School of Medicine (Protocol REC REF numbers: 2017-017 and 2018-099), and the Uganda National Council for Science and Technology (Protocol #s: SS4378 and HS 2466). All caregivers gave written informed consent, and children provided assent or consent for study participation.

### 2.4. Outcome Definitions: Developmental Disorder, Functional Impairment, and Resiliency

Five separate caregiver reported indices were scored following instructions provided in the manual of the *Behavioral Assessment System for Children*, Third Edition (BASC-3) [25]. Each was analyzed as a continuous outcome variable (z-scores) that quantified the children’s propensity to develop behavioral or emotional functioning disorders, be functionally impaired, or exhibit a resilient profile relative to age-matched Ugandan CHUU. The BASC-3 items were forward and back translated for the study setting with adaptations made to preserve meaning in the local context as previously described [26]. Cronbach’s alpha was calculated as measure of reliability for each tool, and the values for respective outcomes ranged from a low of 0.81 to a high of 0.94, demonstrating that the items included within the outcome measures had good to excellent consistency in measuring underlying constructs. Snacks were given to all participants prior to each interview to reduce the distracting impact of hunger/fatigue on their responses.

Autism Spectrum Disorder (ASD) Probability Index: Defined per response to 13 (if child < 11 years) or 18 (if child ≥ 11 years) questionnaire items where caregivers rated the severity of withdrawal, atypical behaviors, impaired communication, leadership aversion, and poor social skills in their children. The ASD probability index score is positively correlated with behavioral ratings in children with clinically diagnosed autism and indicates a greater propensity of exhibiting unusual behaviors, including problems developing/maintaining peer relationships.

Attention Deficit or Hyperactivity Disorder (ADHD) Probability Index: Defined per summation of 11(if child < 11 years) or 9 (if child ≥ 11 years) questionnaire items that positively correlate with behavioral ratings for children clinically diagnosed with ADHD. Higher scores are suggestive of difficulty in academic settings or completion of tasks requiring attention due to inability to sustain focus, plan tasks, make decisions, or moderate activity level.

Emotional Behavioral Disorder (EBD) Probability Index: Defined per summation of 30 (if child < 11 years) or 20 (if child ≥ 11 years) items that positively correlate with behavioral scores obtained in children identified within educational settings as having emotional/behavioral disturbance or disability. Higher scores suggest greater likelihood of exhibiting disruptive, atypical, and antisocial behaviors or high levels of negative emotions like anger, sadness, and pessimism—behavioral traits that contribute to strain in peer and adult relationships.

Functional Impairment (FI): This index was measured using a total of 44 items that capture the level of difficulty children have with engaging in successful or appropriate behaviors during interactions with others, in the execution of age-appropriate tasks, in the regulation of mood, and success with school-related tasks. Elevated scores suggest difficulty responding appropriately in everyday social interactions in a variety of settings.

Resiliency Indices: This index was defined using nine items gauging the overall capacity to engage in adaptive behaviors like coping with change, recovering from setbacks, and problem solving in the face of adversity. High scores suggest greater likelihood of exhibiting a resilient trajectory in the short-term and correlate with positive mental health.

### 2.5. Primary Determinant: In Utero/Peripartum ART (IPA) Exposure

As previously described [26], IPA exposure was established from medical records: namely, the mother’s ART treatment card and antenatal or early-infant diagnosis registers for CPHIV and CHEU. CHEU and CPHIV were exposed to one of four IPA types: (1) no IPA, (2) intrapartum prophylactic single-dose nevirapine (SdNVP) ± zidovudine (AZT), (3) intrapartum prophylactic SdNVP + AZT plus Lamivudine (3TC), i.e., SdNVP + AZT + 3TC, and (4) combination ART (cART, including at least two antiretroviral drug classes), and these children were compared to CHUU (reference).

### 2.6. Other Measures

Current ART Regimen: The current ART regimen of mothers in the CPHIV and CHEU subsamples was abstracted from their medical records at enrolment into the study. Their current ART regimen was in four categories: namely, (1) NRTI (Abacavir or TDF inclusive), (2) NNRTI (EFV and NVP), (3) protease inhibitors (Ritonavir, Atazanivar, Lopinavir/Kaletra), and (4) no current ART/unknown.

Sociodemographic Factors: Biological sex was defined as male vs. female; chronologic age and formal education completed (in years) and developmental stage—i.e., preadolescent vs. adolescent (<11 vs. ≥11 years)—were defined for caregivers and dependent children.

Caregiving Context: The severity of acute caregiver psychosocial stress over the past month was assessed using the perceived stress scale which ranges from 0 (no stress) to 40 (highest stress) [27,28]. Lifetime stress—i.e., sum of occurrence for 13 adverse experiences over the life course—was assessed using the stressful life events questionnaire [29]. Their subjective social standing (lowest = 1 to highest = 10) was assessed using the MacArthur scale of subjective social standing [30]. Their functioning within the caregiving role (lowest = 1 to highest = 120) was measured using an adapted version of the Barkin index of maternal functioning scale [31,32]. Caregiver depressive and anxiety symptoms were measured using 15 and 10 items, respectively, from the Hopkins Symptom Checklist-25 [33,34]. As in our prior studies, social support was measured as the summed score of eight questions in which caregivers expressed agreement or not with statements about their ability to access wanted emotional, monetary, and physical support resources [35].

### 2.7. Statistical Analyses

This secondary analysis included four IPA exposure groups that were compared with CHUU as the primary determinant; hence we estimated the minimum detectable effect size with 80% power using a two-sided test of significance at alpha = 0.05. The minimum detectable effect size varied according to the available number of children within IPA groups in the cohort. Specifically, compared to the 251 CHUU, this study has 80% power with 95% confidence to detect effect sizes of (a) ≥0.3 among the 250 children without IPA exposure, (b) ≥0.4 among the 96 children exposed to SdNVP ± AZT, and (c) ≥0.5 in all IPA categories as each includes ≥75 children per group.

Analysis of variance for continuous measures and chi-square tests for categorical variables were used to summarize differences according to the type of early-life ART exposure and, for CPHIV, according to the developmental stage. Multivariable repeated measures linear regression models were implemented using the generalized estimating equations approach to quantify IPA-related standardized mean differences (SMD) in age standardized outcome measures over 12 months’ follow-up via SAS PROC GENMOD. In all models, confounders such as caregivers’ age, sex, socioeconomic status, depression, and functioning in caregiving role stress were adjusted for based on subject matter knowledge. The random effect of the caregiver was included in all models to account for nesting of children within households. Time was entered as a class variable to model potentially non-linear patterns, and interactions between time and IPA-type were examined to assess potential variation in respective outcomes over 12 months’ observation according to IPA type. In the absence of IPA by time interaction, time averaged associations between repeated assessments of disorders over 12 months were presented. Since outcomes were standardized by age, standardized mean differences (SMD) were estimated—a measure comparable to Cohen’s effect size and thus permitting the determination of clinical importance. Per prior precedent based on studies of quality of life, |SMD| of ≥0.33 were deemed clinically important for child growth which has wide-ranging lifelong implications [36]. Hence, |SMD| < 0.33, 0.33 ≤ |SMD| < 0.50 and |SMD| ≥ 0.50 were considered to be of small to modest, moderate, and large clinical importance, respectively.

To explore the perinatal developmental stage as a potential moderator of IPA associated variations in respective outcomes over 12 months, IPA by developmental stage interaction was introduced in multivariable models. Potential heterogeneity was indicated when *p*-value for the interaction was <0.1. In that case, the results for IPA-related SMD in developmental disorders were presented separately for preadolescent vs. adolescent children. In the absence of heterogeneity, the results were presented for the overall sample. All analyses were performed with SAS version 9.4 (SAS Institute, Inc., Cary, NC, USA). All hypothesis tests were two-sided at alpha = 0.05.

## 3. Results

A sample of 748 children (with complete data for this analysis) of average age 11.4 (SD = 3.7, range 6–18) years old, including 497 born to PWLWH and 251 CHUU, were enrolled. Of the HIV-exposed sample (CPHIV and CHEU), 96 (19.3%), 75 (15.1%), 76 (15.3%), and 250 (50.3%), respectively, were exposed to SdNVP ± AZT, SdNVP + AZT + 3TC, in utero cART, and no ARV in utero/peripartum.

Across the early-life ART exposure groups at baseline, children were similar with respect to distributions of age, sex, birthweight, birth APGAR scores, and current scores on ADHD and EBD probability indices. However, the average scores on ASD, functional impairment, and resiliency metrics at baseline differed across early-life ART exposure groups with the lowest average scores evident among children with SdNVP + AZT + 3TC exposure in utero/peripartum (all *p* < 0.05). The caregivers of the enrolled children were similar across IPA exposure groups with respect to age, biological sex, depressive symptoms, and overall caregiver functioning at enrolment but differed with respect to adverse life-time experiences, and the rating of their social standing in the community (Table 1).

### 3.1. Child Immunologic and HIV Treatment Parameters among CPHIV

Among the 251 CPHIV enrolled, there was no difference in mean scores in the respective developmental disorder probability indices when comparing preadolescent to adolescent subsamples, but the immunologic history, age at cART initiation, and current ART regimen differed. Specifically, the absolute CD4 levels and mean age at child cART initiation were lower and the virologic suppression rates were much higher in preadolescent compared to adolescent CPHIV. The proportion who had ever had an AIDS defining morbidity was similar for preadolescent and adolescent CPHIV. However, current NRTI regimens were applicable only to adolescent CPHIV with most of their preadolescent CPHIV peers being on the current NNRTI regimen (Table 2).

### 3.2. Early-Life ART Exposure Type and Outcomes at 6–18 Years Old: Comparing CHEU vs. CHUU

Relative to CHUU, the disorder probability, resiliency index, and functional impairment scores were similar for CHEU exposed to cART in the peripartum period and for CHEU with no ARV exposure by 6–18 years. Similarly, no difference was apparent in age-standardized ADHD (SMD = −0.19; 95% CI: −0.52, 0.15), ASD (SMD = −0.08; 95% CI: −0.37, 0.22), and EBD (SMD = 0.08, 95% CI: −0.23, 0.43) probability index scores at 6–18 years old for CHEU exposed to peripartum SdNVP±AZT vs. CHUU. However, peripartum SdNVP ± AZT exposure among CHEU was associated with higher resiliency scores (SMD = 0.51, 95% CI: 0.14, 0.87) with an observed difference relative to CHUU of large clinical importance at 6–18 years.

In addition, CHEU with peripartum SdNVP+AZT+3TC exposure exhibited lower resiliency (SMD = −0.25, 95% CI: −0.49, −0.01), and elevated ADHD (SMD = 0.41, 95% CI: 0.12, 0.70), ASD (SMD = 0.40, 95% CI: 0.19, 0.61), and EBD (SMD = 0.32, 95% CI: 0.08, 0.56) probability and functional impairment (SMD = 0.39, 95% CI: 0.18, 0.61) scores at 6–18 years old. The SdNVP + AZT + 3TC associated elevation in respective outcomes was also of moderate clinical importance (Table 3).

### 3.3. Early ART Exposure Type and Outcomes at 6–18 Years Old: Comparing CPHIV vs. CHUU

Among CPHIV exposed to no antiretroviral drugs, CPHIV exposed to SdNVP ± AZT and CPHIV exposed to cART in the peripartum period did not differ with respect to probability index scores for respective developmental disorders at 6–18 years old relative to CHUU. Further, there were no differences in respective developmental outcome indicators at 6–18 years for the comparison of CPHIV exposed to SdNVP ± AZT or CPHIV exposed to cART to CPHIV peers who were not exposed to any antiretroviral drugs in their early life. However, at 6–18 years old, peripartum exposure to SdNVP + AZT + 3TC was associated with elevated time averaged ASD (SMD = 0.47, 95% CI: 0.09, 0.86) and EBD (SMD = 0.27, 95% CI: 0.01, 0.53) probability index scores and with lower resiliency (SMD = −0.36, 95% CI: −0.66, −0.07) over a 12 month period compared to the CHUU. Relative to CPHIV without any IPA exposure, the SdNVP + AZT + 3TC associated worse z-scores in developmental disorders persisted despite adjustment for HIV-treatment factors (Table 4).

### 3.4. Child Factors and Relationship to Developmental Outcomes

Per unit increase in birth APGAR score, the ASD probability index decreased whereas the resiliency index increased significantly at 6–18 years old for all children, including CPHIV. Likewise, higher birthweight was inversely associated with ADHD (SMD = −0.19, 95% CI: −0.39, −0.00) and ASD (SMD = −0.18, 95% CI: −0.33, −0.03) probability indices at 6–18 years of age (Appendix A), although this association was generally absent in analyses restricted to CPHIV (Appendix A). Except for the EBD (SMD = −0.22, 95% CI: −0.37, −0.07) probability index scores that were lower in girls vs. boys, child sex was not a significant predictor of outcomes at 6 to 18 years (Appendix A).

### 3.5. Caregiving Environment and Developmental Disorders at 6–18 Years Old

Children whose caregivers reported lower vs. highest levels of functioning in the caregiving role had comparable resiliency index scores whereas scores on ADHD (SMD = 0.13 to 0.32, 95% CI: −0.03, −0.49), EBD (SMD = 0.35 to 0.45, 95% CI: 0.18, 0.60), and ASD (SMD = 0.17 to 0.29, 95% CI: 0.02, 0.44) were minorly to moderately elevated (Appendix A). Similarly, children whose caregivers reported high vs. low adversity level over their life course had a small likelihood for ADHD, ASD, EBD, and FI over the study period. Caregiver depression was largely not associated with respective outcomes, although lack of depression predicted a lower EBD probability index over the study period (Appendix A).

### 3.6. HIV Treatment Factors and Outcomes at 6–18 Years Old among CPHIV

Among CPHIV on cART, the outcomes did not differ according to the type of current cART regimen. However, the ASD probability (SMD = 0.47, 95% CI: 0.01, 0.93) and FI (SMD = 0.58, 95% CI: 0.07, 1.09) scores at 6–18 years were elevated for CPHIV not on treatment at enrolment vs. peers on NNRTI-based cART. With few exceptions, neither CD4 nadir nor child age at cART initiation was associated with respective outcomes (Appendix A).

## 4. Discussion

Consistent with our study hypothesis, the association of IPA type with developmental disorder, functional impairment, and resiliency outcomes at 6–18 years of life varied according to regimen. Specifically, we observed that relative to CHUU, the disorder probability, functional impairment, and resiliency index scores at 6–18 years old were comparable for HIV-exposed children without coincident IPA exposure (i.e., HIV natural history cohort), and HIV-exposed children exposed to cART and SdNVP ± AZT in the peripartum period. One exception to the general trend of no association was that peripartum SdNVP ± AZT exposure predicted higher resiliency scores for CHEU vs. CHUU at 6–18 years old. These findings were consistent with the previously reported minimal risk of acute and long-term neurological outcomes from in utero AZT exposure in CHEU compared to CHUU [37,38]. Considering the increasing and unavoidable use of cART among PWLWH, this observation suggested that neither intrapartum SdNVP ± AZT nor peripartum cART exposure use among PWLWH elevated developmental disorder probability and functional impairment later in the life course for their newborns. Replication of these observations will be important in other samples and, if confirmed, will be reassuring for millions of children that will continue to be exposed to cART early in the peripartum period. The large, programmatic investment of the National Institutes of Health in epidemiologic studies is optimally positioned to illuminate, qualify, and contextualize our observations with a specifically designed set of investigations to address this question in CHEU populations around the world [39].

Of concern is our observation that peripartum SdNVP + AZT + 3TC exposure was associated with lower resiliency and elevated ADHD, ASD, and EBD probability and functional impairment index score in comparison with the community controls at 6–18 years old. This regimen’s associated risk of respective outcomes was of moderate to high clinical importance, suggesting that peripartum exposure to SdNVP + AZT + 3TC may disrupt neural mechanisms underlying neurobehavioral adaptation in exposed children. Although no association was made between developmental disorders and perinatal ART exposure, two cross-sectional investigations among school-age and adolescent CPHIV from Uganda have identified behavioral, psychiatric, sensory, and other behavioral disorders as a public health problem that often co-occurs with psychiatric disorder among children born to PWLWH [17,20]. Additionally, CHEU vs. CHUU status was associated with three times greater odds of any developmental disorder diagnosis in a diverse sample of Canadian children, but the study team found no evidence that the elevated risk of any neurodevelopmental disorder diagnosis was differentially elevated for any ART regimen [12]. Support for the adverse association noted in this sample for the 3TC-inclusive ART regimen comes from two epidemiologic studies. For the first, researchers in a study of evoked potentials in 74 infants at 40 weeks found subclinical evidence of worse function in the lower brain stem regions for those AZT + 3TC-exposed relative to control infants [40]. For the second, we have previously reported that early SdNVP + AZT + 3TC exposure led to socioemotional problems among 6–10 years old children from Uganda [26].

The vulnerability of CHEU and CPHIV to adverse cognitive and behavioral outcomes is an emergent public health concern [16], and ours is among a handful of studies investigating IPA exposure type and long-term functional impairments and developmental disorders [20,21]. The oldest children in this sample were born in the year 2000, a time when PMTCT programs for pregnant women in Uganda were limited in scope and PWLWH received ART for their own health subject to meeting immune suppression thresholds. A higher proportion of the youngest children (born in the year 2012) were within the Option B+ era during which PWLWH began accessing lifelong cART regardless of their CD4 cell count. The type of PMTCT interventions (if available) in PWLWH evolved substantially with scientific knowledge and programmatic scale-up in resource-limited settings between the years 2000 and 2012. Accordingly, 50% of HIV-affected children were not exposed to any IPA, and the remaining 50% were exposed to SdNVP ± AZT, SdNVP + AZT + 3TC, or cART depending on the applicable standard of care for PMTCT during their pregnancy/birth. This substantial variation in IPA exposure increases the statistical efficiency in relating IPA to developmental and neurobehavioral disorders at 6–18 years. Of note, behavioral dysfunction is more reliably recognizable to caregivers at this life stage, which also coincides with peak periods of myelination and gliogenesis in human brain development [6,41,42].

Information arising from in vitro, animal, and ART pharmacologic studies when superimposed on the expected neural changes across the human life course taken together affirms the scientific premise that variations in ART exposure type may have delayed and long-lasting neurobehavioral consequences in HIV-affected human populations. The extended period of dynamic changes in human developmental processes provides several windows of opportunity beginning from neurulation and early neurogenesis which occur in utero, to synaptogenesis, gliogenesis, myelination, and synaptic pruning which continue throughout childhood and into early adulthood [6] for variations in the ART regimen to influence the behavioral trajectory potentially manifesting as disorders and functional impairment observed in this study. Specifically, preclinical in vitro and animal studies [6,43] have linked exposure to antiretroviral drugs with lower maturation, survival, and worse myelination in oligodendrocytes—the neural cell types that maintain axonal integrity and facilitate signal transduction [44]. Evidence from studies in in utero ART exposure, including AZT+3TC in mice, has been linked to a range of atypical behaviors in the adolescent period of mouse life and suggested the need for appropriate vigilance regarding the possible delayed impact of early ART manifesting as abnormal social, emotional, and behavioral adjustment in perinatal ART-exposed humans [45,46,47,48].

Indeed, molecular pharmacologic studies established that the ability of antiretroviral drugs to cross the blood–brain barrier increases when ARVs are used in combination [49]. As the standard of care evolved rapidly across the HIV eras, an underlying driver of the rapid change to 3TC-inclusive formulations was to improve the central nervous system penetration of respective drugs [50]. The observation of a clinically large increase in respective disorder probabilities and functional impairment scores for intrapartum SdNVP + AZT + 3TC relative to (a) no ARV exposure at all (natural history) and (b) simpler intrapartum SdNVP ± AZT combinations was consistent with the expected higher CNS penetration effectiveness noted in pharmacologic studies [49,50,51]. The findings from this study demonstrated that variations in constituent ART drugs may have an impact on developmental phenotype later in the life-course. However, the logical expectation that in utero cART exposure—just like 3TC inclusive regimen—will predict worse disorder probability was not confirmed by data arising from this sample. Hence, future studies are needed to elucidate proximate mechanisms underlying the qualitative difference in the direction of association observed for child intrapartum exposure to SdNVP + AZT + 3TC relative to cART by PWLWH.

### Limitations, Strengths, and Future Directions

The absence of randomization in IPA exposure types and our inability to gain greater resolution regarding the timing of cART in the current pregnancy constrains worked together to limit causal attribution. In addition, the outcomes were defined using standardized caregiver reported questionnaire items with acceptable to high reliability for behavioral manifestations consistent with respective disorders [25]. However, we did not implement confirmatory clinical diagnosis of the respective outcomes based on developmental disorders with the DSM-5 or other gold-standard instruments. Future studies that include clinically confirmed diagnosis will be important to definitively identify children with developmental disorders. Despite these limitations, the use of prospective cohort design, a large sample size including demographically matched CHUU, and a large variation in IPA exposure types were design strengths that allowed us to delineate differences in respective outcomes according to in utero HIV exposure with/without IPA exposure. Other strengths that should inspire high confidence in the results presented include the objective determination of IPA exposures and the ability to exclude critical competing risk factors (e.g., low birth weight, birth APGAR), variations in caregiving quality/environment, caregiver emotional wellbeing and socioeconomic status as alternate explanations for the observed findings.

## 5. Conclusions

We found no evidence that intrapartum SdNVP ± AZT and peripartum exposure to cART were adversely associated with developmental outcomes/resiliency at 6–18 years. However, prenatal exposure to SdNVP + AZT + 3TC predicted a large and clinically important elevation in the probability indices for certain developmental disorders and functional impairment assessed in late childhood and adolescence. Specifically, designed future studies will be important to confirm or refute the data arising from this sample. If confirmed, the lack of association between cART exposure with developmental disorder indices at 6–18 years is important endorsement of the long-term safety of peripartum cART with respect to functional outcomes and developmental disorders later in the life course for children born to PWLWH. These findings underscore the importance of implementing specifically designed mechanistic and clinical epidemiologic studies to understand proximate mechanisms and identify intervention targets to improve the developmental trajectory of HIV- and ART-exposed children of PWLWH.

## Figures and Tables

**Table 1 ijerph-19-03725-t001:** Baseline description of children and caregivers in the study base with respect to type of early-life ART exposure.

	SdNVP ± AZT (N = 96)	SdNVP + AZT + 3TC (N = 75)	In utero cART (N = 76)	No ARV (N = 250)	5 HUU, (N = 251)	Group Comparison χ^2^/ANOVA Test
Child Sociodemographic Factors	n (%)	n (%)	n (%)	n (%)	n (%)	
% Female	59 (63.0)	41.0 (55.0)	41.0 (54.0)	120 (49.0)	123 (63.0)	0.1965
	Mean (SD)	Mean (SD)	Mean (SD)	Mean (SD)	Mean (SD)	
Age (years)	11.8 (3.9)	11.9 (3.5)	10.4 (3.5)	11.5 (3.7)	11.4 (3.7)	0.0809
Birth weight (in kg)	3.22 (0.48)	3.29 (0.58)	3.27 (0.43)	3.27 (0.49)	3.29 (0.54)	0.847
Apgar Score	8.4 (0.9)	8.2 (0.8)	8.4 (0.9)	8.3 (0.9)	8.4 (1.04)	0.364
Age- and sex-standardized Child Disorders, Impairment and Resiliency Metrics (in z-scores)						
Attention deficit hyperactivity disorder (ADHD)	−0.2 (1.1)	0.4 (0.8)	0.1 (1.0)	−0.1 (1.1)	0.0 (1.1)	0.421
Autism Spectrum Disorder (ASD)	−0.1 (1.1)	0.5 (0.8)	0.2 (0.9)	−0.1 (1.0)	0.0 (1.0)	0.004
Emotional Behavior Disorder (EBD)	−0.1 (1.1)	0.3 (0.8)	0.0 (0.9)	0.0 (1.0)	0.0 (1.0)	0.0825
Functional Impairment Index	−0.1 (1.3)	0.5 (0.7)	0.1 (0.9)	0.0 (1.0)	0.0 (1.0)	0.003
Resiliency Index	0.4 (1.1)	−0.3 (0.8)	0.0 (1.1)	0.1 (1.0)	0.0 (1.0)	0.002
Caregiver Demographics	n (%)	n (%)	n (%)	n (%)	n (%)	
% Female	75 (86.2)	64 (86.5)	66 (89.2)	207 (88.5)	215 (88.8)	0.9487
	Mean (SD)	Mean (SD)	Mean (SD)	Mean (SD)	Mean (SD)	
Age (years)	38.4 (10.9)	40.7 (11.8)	40.3 (13.0)	39.9 (11.8)	38.2 (10.8)	0.2979
Education (years)	6.6 (3.04)	5.72 (3.13)	6.31 (4.28)	6.02 (3.73)	7.01 (3.83)	0.0183
Depressive Symptom score	9.3 (8.7)	10.3 (8.4)	12.6 (8.9)	11.1 (8.9)	10.7 (8.9)	0.1837
Adverse Lifetime Experiences	1.7 (2.0)	2.8 (3.0)	2.7 (2.5)	2.2 (2.3)	2.0 (2.3)	0.0038
Social Standing McArthur Scale	3.0 (1.3)	3.3 (1.4)	3.8 (1.8)	3.2 (1.4)	3.6 (1.4)	0.0007
Functioning in Caregiving Role	59.5 (10.0)	60.2 (9.2)	58.3 (8.4)	57.3 (8.5)	57.8 (10.5)	0.0922

**Table 2 ijerph-19-03725-t002:** Description of caregiving context, child immunologic and HIV treatment parameters among perinatally HIV-infected children according to developmental stage.

	PreAdolescent N = 99	Adolescent N = 152	t/χ^2^
Caregiver Sociodemographic and psychosocial factors	Mean (SD)	Mean (SD)	*p*-Value
Age (in years)	34.14 (8.73)	39.40 (11.50)	<0.001
Depressive Symptoms	9.51 (7.46)	9.82 (9.53)	0.7837
Adverse lifetime experiences	2.47 (2.41)	1.60 (1.83)	0.0014
Functioning in Caregiving Role	56.57 (9.06)	59.76 (9.03)	0.007
Child Immunologic History	Mean (SD)	Mean (SD)	
Current CD4	1281 (657)	578 (250	<0.001
CD4 NADIR (lowest ever)	762 (490)	481(238)	<0.001
Proportion stably suppressed n (%)	56 (51)	37 (24)	0.01
Peripartum ART Exposure Type	n (%)	n (%)	
CPHIV SdNVP ± AZT	22 (22)	41 (27)	0.82
CPHIV SdNVP + AZT + 3TC	10 (10.1)	17 (11.18)	
CPHIV in utero cART	10 (10.1)	14 (9.21)	
CPHIV no peripartum ARV exposure	57 (57.58)	80 (52.6)	
Estimated age of combination ART Initiation	n (%)	n (%)	
≤6 months	24 (24.0)	15 (9.9)	0.02
>6–18 months	29 (29.0)	62 (41.1)	
>18–48 months	21 (21.0)	22 (14.6)	
>48–60 months	12 (12.0)	25 (16.6)	
>60 months	14 (14.0)	27 (17.9)	
Current combination ART	n (%)	n (%)	
NRTI Regimen	0 (0)	54 (34.8)	<0.001
NNRTI Regimen	66 (66.0)	62 (40.0)	
Protease Inhibitor **	31 (31.0)	33 (22.3)	
None or Unknown current ART	3 (3.0)	6 (3.88)	
AIDS Defining Morbidity *	19 (19.2)	21 (13.9)	0.27
Developmental Disorder Probability Index, Impairment and Resiliency Metrics at 6–18 years	z-score (Mean, SD)	z-score (Mean, SD)	
Attention deficit hyperactivity disorder (ADHD)	−0.22 (1.01)	0.04 (1.07)	0.0578
Emotional Behavior Disorder (EBD)	−0.03 (1.05)	0.02 (1.04)	0.7111
Autism Spectrum Disorder (ASD)	−0.06 (0.94)	0.06 (1.07)	0.3659
Functional Impairment Index	−0.08 (1.06)	0.12 (1.11)	0.1557
Resiliency Index	0.19 (0.93)	0.08 (1.14)	0.3943

*: Includes previous TB, severe malnutrition, or severe immune deficiency; **: PI includes Ritonavir, Atazanivar, or Lopinavir/Kaletra; NNRTI = EFV or NVP; NRTI = Abacavir or TDF inclusive.

**Table 3 ijerph-19-03725-t003:** Early-life ART exposure in relationship to ADHD, ASD, and EBD probability indices and overall resiliency among 6–18 years old HIV-exposed uninfected children relative to community control children from Uganda.

	ADHD Probability Index			ASD Probability Index		EBD Probability Index		Functional Impairment Index		Resiliency Index	
Peripartum ART Exposure Type	n	LSM ± SE	SMD (95% CI)	LSM ± SE	SMD (95% CI)	LSM ± SE	SMD (95% CI)	LSM ± SE	SMD (95% CI)	LSM ± SE	SMD (95% CI)
CHEU SdNVP ± AZT	34	−0.37 ± 0.17	−0.19 (−0.52, 0.15)	−0.24 ± 0.15	−0.08 (−0.37, 0.22)	0.04 ± 0.16	0.08 (−0.23, 0.43)	−0.19 ± 0.19	−0.05 (−0.41, 0.34)	0.65 ± 0.18	**0.51** **(0.14, 0.87)**
CHEU SdNVP + AZT + 3TC	48	0.21 ± 0.11	**0.41** **(0.12, 0.70)**	0.25 ± 0.10	**0.40** **(0.19, 0.61)**	0.28 ± 0.11	**0.32** **(0.08, 0.56)**	0.25 ± 0.10	**0.39** **(0.18, 0.61)**	−0.10 ± 0.11	**−0.25** **(−0.49, −0.01)**
CHEU cART in utero or peripartum	52	−0.11 ± 0.13	0.03 (−0.24, 0.30)	−0.12 ± 0.12	0.03 (−0.21, 0.28)	−0.08 ± 0.12	−0.04 (−0.30, 0.22)	−0.20 ± 0.11	−0.06 (−0.27, 0.19)	0.25 ± 0.13	0.11 (−0.17, 0.38)
CHEU no ARV exposure	104	−0.20 ± 0.09	−0.06 (−0.27, 0.16)	−0.15 ± 0.09	0.01 (−0.18, 0.19)	−0.08 ± 0.09	−0.04 (−0.24, 0.15)	−0.18 ± 0.08	−0.04 (−0.23, 0.15)	0.11 ± 0.08	−0.03 (−0.22, 0.16)
CHUU	249	−0.14 ± 0.06	Ref	−0.16 ± 0.06	Ref	−0.03 ± 0.06	Ref	−0.14 ± 0.06	Ref	0.14 ± 0.06	Ref

Multivariable regression models adjusted for time, child (age, sex), caregiver (education, lifetime adversity, depressive symptoms, caregiver functioning in caregiving role, social support), and early-life child health indicators (birthweight, APGAR score); bolded numbers are statistically significant associations.

**Table 4 ijerph-19-03725-t004:** Early-life ART exposure in relationship to ADHD probability, EBD probability, and overall resiliency at 6–18 years old for children with perinatally acquired HIV infection relative to ARV unexposed peers and children HIV unexposed uninfected from Uganda.

		ASD Probability Index	ADHD Probability Index	EBD Probability Index	Functional Impairment Index	Resiliency Index
	n	SMD (95% CI)	SMD (95% CI)	SMD (95% CI)	SMD (95% CI)	SMD (95% CI)
**Peripartum ART Exposure Type**						
CPHIV SdNVPSdNVP ± AZT	63	0.00 (−0.28, 0.28)	−0.12 (−0.41, 0.18)	0.01 (−0.26, 0.29)	−0.04 (−0.32, 0.27)	0.20 (−0.05, 0.46)
CPHIV SdNVP + AZT + 3TC	27	**0.47 (0.09, 0.86)**	0.23 (−0.03, 0.50)	**0.27 (0.01, 0.53)**	**0.51 (0.26, 0.78)**	**−0.36 (−0.66, −0.07)**
CPHIV in utero cART	24	0.09 (−0.13, 0.30)	0.04 (−0.27, 0.36)	−0.11 (−0.40, 0.18)	0.19 (−0.11, 0.52)	−0.04 (−0.32, 0.24)
CPHIV no peripartum ARV exposure	137	−0.04 (−0.21, 0.12)	−0.06 (−0.24, 0.11)	0.05 (−0.13, 0.23)	0.03 (−0.15, 0.20)	0.08 (−0.10, 0.25)
CHUU	249	Ref	Ref	Ref	**Ref**	**Ref**
**CPHIV Sample Restricted Analyses further Adjusted for HIV-treatment related Factors ***						
SdNVP ± AZT	62	0.09 (−0.31, 0.32)	−0.01 (−0.35, 0.33)	−0.04 (−0.36, 0.29)	−0.00 (−0.34, 0.23)	0.17 (−0.14, 0.48)
SdNVP + AZT + 3TC	26	**0.50 (0.03, 0.80)**	0.32 (−0.01, 0.65)	0.27 (−0.02, 0.56)	**0.46 (0.16, 0.76)**	**−0.38 (−0.72, −0.04)**
in utero cART	24	0.26 (−0.04, 0.56)	0.12 (−0.20, 0.46)	−0.09 (−0.45, 0.27)	0.30 (−0.07, 0.66)	0.01 (−0.39, 0.40)
No IPA	130	Ref	Ref	Ref	Ref	

In addition to variables above, multivariable regression model adjusted for time and the following child factors (birth APGAR, birthweight, biological sex, adolescent vs. preadolescent developmental stage) and caregiver factors at enrolment: education, lifetime adversity, depressive symptoms, caregiver functioning in caregiving role, social support. * HIV-treatment related parameters include current ART regimen, CD4 nadir, age at ART initiation (<18 months vs. ≥18 months). Bolded numbers are statistically significant associations.

## Data Availability

The data presented in this study are available on request from the corresponding author subject to data sharing agreements.

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
