# Peer review of "Developmental Disorder Probability Scores at 6–18 Years Old in Relation to In-Utero/Peripartum Antiretroviral Drug Exposure among Ugandan Children"

_ijerph, 2022, doi:10.3390/ijerph19063725_

Round 1

Reviewer 1 Report

The paper is valuable and perfect as it is. It makes 2 major points:

1- 3TC based regimens - we need to be open that there could be increased ASD type phonemonon later

2-cART does not have this association

One very minor suggestion : ? delete "of regimen" in line 373

The limitations section at the end is quite long.

Author Response

Response: We thank Reviewer #1 for appreciating the essence of our study and correctly distilling the important inferences from our work.

1. One very minor suggestion: ? delete "of regimen" in line 373

Response: Thank you for pointing out this area that required clarification.  We have now deleted rephrased to improve the communication. 

2. The limitations section at the end is quite long.

Response:  We do not disagree with the reviewer.  We have read through this section again being careful to balance the emerging inferences with potential limitations that must be kept in mind keeping our observational study design in mind.

Reviewer 2 Report

The manuscript provides the research results of a study that evaluated antiretroviral exposure in utero/peripartum and the probability of attention deficit hyperactivity disorder, autism spectrum disorder, and functional impairment. However, I have some comments that will improve the relevance of the study:

  1. Introduction:

- The authors should raise the problem of the situation of antiretroviral exposure and developmental disorders not only globally but also on the African continent, especially in Uganda.

- It is important to describe some similar studies worldwide. If they do not exist, they should indicate it.

- Although the hypothesis is present, the study's primary objective needs to be added.

Methods:

- They do not indicate the calculation of the sample size, nor do they indicate whether the sample was chosen randomly or not.

- In describing the assessment instruments, the authors do not indicate the validity and reliability of the instruments in the Ugandan population. It is crucial to describe the instruments in greater detail and why these tests were chosen and not others, such as the Wechsler scales, which have more excellent reliability and validity worldwide to assess cognitive functioning in children. The reason for not applying these tests should be described.

- How did you verify that there were no information biases on the part of the informants in the behavioral questionnaires? How to verify whether or not the child exhibits the behavior described by the mother?

- The design of the mentioned study corresponds to a prospective cohort. However, they used a multivariate model rather than a cross-sectional study. I suggest performing a multivariate model analysis that considers the time variable as a GEE or a mixed-effects model. Average the different results over time to induce error in the association between variables.

Results:

- The authors do not indicate if the results are normally distributed. This should be pointed out and show the result of the Shapiro Wilk test or the Kolmogorov Smirnov test to justify the use of parametric tests in the comparison

- The authors assume that the data are normal and report the mean and standard deviation. However, they must describe the median and interquartile ranges of the data in order to observe the central tendency and dispersion of the data.

- If the data is not normally distributed, non-parametric tests should be applied for the bivariate analysis of continuous data.

- The multivariate regression model should consider the measurements over time of the test results. It is important to indicate the goodness of fit of the regression models and the r2 or coefficient of determination of the model.

Discussion:

- With the new statistical analysis suggested, the discussion should be revised.

- They must contrast the results obtained with other similar studies.

- In the study's limitations, they must point out the lack of use of validated cognitive tests that directly evaluate the behavior or response of the child and not through a questionnaire applied to a third party or the parents.

- The authors should make suggestions to specialists and decision-makers about the results found.

Round 2

Reviewer 2 Report

I reviewed the new version and appreciate the responses from the authors. I agree to the publication of the manuscript with the changes incorporated